# Dietary Antioxidants Significantly Attenuate Hyperoxia-Induced Acute Inflammatory Lung Injury by Enhancing Macrophage Function via Reducing the Accumulation of Airway HMGB1

**DOI:** 10.3390/ijms21030977

**Published:** 2020-02-01

**Authors:** Vivek Patel, Katelyn Dial, Jiaqi Wu, Alex G. Gauthier, Wenjun Wu, Mosi Lin, Michael G. Espey, Douglas D. Thomas, Charles R. Ashby, Lin L. Mantell

**Affiliations:** 1Department of Pharmaceutical Sciences, College of Pharmacy, St. John’s University Queens, Queens, NY 11439, USA; patelvivek85@gmail.com (V.P.); katelyn.dial12@my.stjohns.edu (K.D.); jiaqi.wu16@stjohns.edu (J.W.); alex.gauthier15@my.stjohns.edu (A.G.G.); wwjcpu@gmail.com (W.W.); lms.47ace@gmail.com (M.L.);; 2National Cancer Institute, Bethesda, MD 20892-9747, USA; michael.espey@nih.gov; 3Department of Medicinal Chemistry and Pharmacognosy, University of Illinois at Chicago, Chicago, IL 60612, USA; ddthomas@uic.edu; 4The Feinstein Institute for Medical Research, Northwell Health System, Manhasset, NY 11030, USA

**Keywords:** ascorbic acid, sulforaphane, HALI, ARDS, HMGB1, Nrf2, dietary antioxidants, macrophages, oxidative stress, phagocytosis

## Abstract

Mechanical ventilation with hyperoxia is the major supportive measure to treat patients with acute lung injury and acute respiratory distress syndrome (ARDS). However, prolonged exposure to hyperoxia can induce oxidative inflammatory lung injury. Previously, we have shown that high levels of airway high-mobility group box 1 protein (HMGB1) mediate hyperoxia-induced acute lung injury (HALI). Using both ascorbic acid (AA, also known as vitamin C) and sulforaphane (SFN), an inducer of nuclear factor (erythroid-derived 2)-like 2 (Nrf2), we tested the hypothesis that dietary antioxidants can mitigate HALI by ameliorating HMGB1-compromised macrophage function in phagocytosis by attenuating hyperoxia-induced extracellular HMGB1 accumulation. Our results indicated that SFN, which has been shown to attenute HALI in mice exposed to hyperoxia, dose-dependently restored hyperoxia-compromised macrophage function in phagocytosis (75.9 ± 3.5% in 0.33 µM SFN versus 50.7 ± 1.8% in dimethyl sulfoxide (DMSO) control, *p* < 0.05) by reducing oxidative stress and HMGB1 release from cultured macrophages (47.7 ± 14.7% in 0.33 µM SFN versus 93.1 ± 14.6% in DMSO control, *p* < 0.05). Previously, we have shown that AA enhances hyperoxic macrophage functions by reducing hyperoxia-induced HMGB1 release. Using a mouse model of HALI, we determined the effects of AA on hyperoxia-induced inflammatory lung injury. The *i.p.* administration of 50 mg/kg of AA to mice exposed to 72 h of ≥98% O_2_ significantly decreased hyperoxia-induced oxidative and nitrosative stress in mouse lungs. There was a significant decrease in the levels of airway HMGB1 (43.3 ± 12.2% in 50 mg/kg AA versus 96.7 ± 9.39% in hyperoxic control, *p* < 0.05), leukocyte infiltration (60.39 ± 4.137% leukocytes numbers in 50 mg/kg AA versus 100 ± 5.82% in hyperoxic control, *p* < 0.05) and improved lung integrity in mice treated with AA. Our study is the first to report that the dietary antioxidants, ascorbic acid and sulforaphane, ameliorate HALI and attenuate hyperoxia-induced macrophage dysfunction through an HMGB1-mediated pathway. Thus, dietary antioxidants could be used as potential treatments for oxidative-stress-induced acute inflammatory lung injury in patients receiving mechanical ventilation.

## 1. Introduction

Globally, acute respiratory distress syndrome (ARDS) accounts for 10% of intensive care unit admissions and results in 3 million deaths annually [1]. The clinical management of ARDS primarily involves the use of mechanical ventilation with supraphysiological concentrations of oxygen (i.e., hyperoxia) [1]. Although mechanical ventilation is a life-saving intervention for patients with respiratory distress or failure, it can exacerbate injury in previously damaged lungs [2,3], resulting in significantly high morbidity and mortality rates in patients on ventilation [4,5]. Prolonged exposure to hyperoxia can cause hyperoxia-induced acute lung injury (HALI). The HALI is characterized by diffused alveolar epithelial damage, disruption of the alveolar capillary barrier with increased pulmonary microvascular permeability, significant pulmonary edema and infiltration of a large number of leukocytes into the lung [6,7,8,9,10].

The prolonged exposure to hyperoxia can induce excessive generation of reactive oxygen species (ROS), particularly superoxide, by both the mitochondrial electron transport chain and NADPH oxidases [2,9,11]. The excessive production of ROS can damage lung cells and produce pulmonary dysfunction [2,9,12]. For example, hyperoxia-induced oxidative stress adversely affects alveolar macrophage function in phagocytosis, leading to decreased clearance of invading pathogens including bacteria [13,14,15,16]. Previously, we reported that hyperoxia-impaired macrophage function in phagocytosis is at least partially due to actin oxidation, causing cytoskeleton disorganization, and the antioxidants, superoxide dismutase (SOD) and procysteine, attenuate this damage [14]. The goal of this study is to determine if dietary antioxidants can reduce HALI via enhancing the macrophage functions compromised by prolonged exposure to hyperoxia.

Although lung damage in HALI can result from direct damage caused by excessive ROS, it may also be mediated by the airway accumulation of high-mobility group box 1 protein (HMGB1), a damage-associated molecular pattern (DAMP) protein [6]. Extracellular HMGB1 can not only impair macrophage functions [17], but also exacerbate hyperoxia-induced inflammatory lung injury [6]. The administration of an anti-HMGB1 antibody in a mouse model of HALI significantly reduced lung injury and neutrophil infiltration in the lungs [6]. The impairment of macrophage phagocytosis by HMGB1 can also decrease the clearance of apoptotic neutrophils [18], and result in inflammatory lung injury as observed in animals administrated with HMGB1 [19,20]. However, it is unclear if antioxidants are efficacious in attenuating HALI by enhancing HMGB1-impaired macrophage phagocytic function.

Nuclear factor erythroid 2-related factor 2 (Nrf2) is a transcription factor that regulates the antioxidant response element (ARE)-mediated transcriptional activation of several endogenous phase 2 detoxifying and antioxidant enzymes [21,22]. Nrf2 has been shown to provide protection against hyperoxia-induced lung injury [21,23]. Mice exposed to 95%–98% O_2_ with a site-directed mutation in the Nrf2 gene had an exacerbation of pulmonary hyper-permeability, neutrophilic inflammation and epithelial injury compared to wild type mice [21]. Furthermore, wild type mice treated with the Nrf2 activator, sulforaphane (SFN), orally or in their diet preceding hyperoxia exposure, had reduced hyperoxia-induced pulmonary injury and oxidation indices compared to wild type mice [22]. However, it remains unknown as to whether the attenuation of HALI by supplemental SFN is due to attenuated hyperoxia-impaired macrophage phagocytosis produced by inhibiting hyperoxia-induced HMGB1 release into the extracellular milieu.

Ascorbic acid (AA) is a dietary supplement and an essential vitamin that scavenges ROS by donating electrons to produce a more reductive environment [24,25,26,27]. Previously, we have reported that the addition of AA to cultured macrophages decreases hyperoxia-impaired macrophage phagocytosis by decreasing extracellular HMGB1 accumulation [28]. In this study, we tested the hypothesis that AA supplementation effectively attenuates hyperoxia-induced HMGB1 accumulation in the airways, resulting in a better clearance of leukocytes in the lung and a substantially reduced extent of HALI.

In this study, we also tested whether SFN can attenuate hyperoxia-compromised macrophage phagocytosis and inhibit hyperoxia-induced extracellular accumulation using the established phagocytosis assay and Western blot analysis for HMGB1 in cultured macrophages as described in previously published studies [6,16,17]. In addition, the effects of AA on inflammatory response, HALI, and on the airway accumulation of HMGB1 were determined in our established mouse model of HALI as described [6]. Figure 1 below illustrates the study design of these proposed experiments. Our results indicated that SFN significantly (1) improved hyperoxia-impaired macrophage function and (2) decreased the extracellular accumulation of HMGB1. The administration of AA to hyperoxia-exposed mice significantly improved clinically relevant outcomes such as lung injury and reduced excessive leukocyte infiltration by decreasing the airway accumulation of HMGB1. Thus, these data indicate that the antioxidants SFN and AA are efficacious in reducing hyperoxia-induced acute inflammatory lung injury by attenuating HMGB1-mediated macrophage dysfunction.

## 2. Results

### 2.1. Sulforaphane Significantly Attenuates Hyperoxia-Induced Dysfunction of Macrophage Phagocytosis

Hyperoxia impairs macrophage function in phagocytosis, thereby diminishing the host defense in clearing apoptotic neutrophils, resulting in pronounced accumulation of leukocytes in the airways [14,18,28,29]. To determine if SFN, which was recently shown to-attenuate HALI [22], can enhance macrophage functions, the phagocytic function of macrophages was determined using cultured macrophages. Prolonged exposure to hyperoxia significantly impaired the phagocytic function of RAW 264.7 cells (49.1 ± 4.1% in hyperoxia versus 99 ± 2.1% in room air, *p* < 0.05, Figure 2A). The incubation of RAW 264.7 cells with SFN significantly increased macrophage phagocytic function in a concentration-dependent manner under hyperoxic conditions (68.5 ± 2.6% in the 0.11 µM group, 75.9 ± 3.5% in the 0.33 µM group, and 87.5 ± 2.9% in the 1 µM group compared to 50.7 ± 1.8% in the vehicle control group, *p* < 0.05, Figure 2A). Importantly, SFN’s restorative effect of hyperoxia-compromised phagocytic function was also observed in primary macrophages. Under hyperoxic conditions, bone-marrow derived macrophages (BMDMs) had a significant impairment in phagocytic function when compared to the room air control group (54.8 ± 0.79% versus 100 ± 0.61%, *p* < 0.05; Figure 2B). The prolonged exposure of BMDMs to hyperoxia in the presence of SFN (0.11, 0.33 or 1 µM) significantly increased macrophage phagocytic function in a concentration-dependent manner (65.3 ± 1.3% in the 0.11 µM group, 75.9 ± 2.8% in the 0.33 µM group, and 83.9 ± 2.7% in the 1 µM group compared to 56.6 ± 1.7% in the vehicle control group, *p* < 0.05, Figure 2B). These results suggest that SFN can attenuate hyperoxia-compromised phagocytosis function in both transformed macrophages as well as primary macrophages.

### 2.2. Sulforaphane Significantly Attenuates Hyperoxia-Induced Oxidative Stress

Nrf2 has been reported to have a prophylactic effect in animals model of ALI induced by hyperoxia, cigarette smoke, and oleic acid [21,30,31,32]. To determine whether SFN mitigates HALI by reducing hyperoxia-induced oxidative stress, macrophages were cultured under hyperoxic conditions and incubated with SFN. Intracellular ROS levels were significantly increased in macrophages exposed to hyperoxia compared to those exposed to room air (3.1 ± 0.065 × 10^4^ versus 2.2 ± 0.025 × 10^4^ AU, *p* < 0.05, Figure 3). SFN (0.11, 0.33 and 1 µM) produced a significant decrease in ROS levels in macrophages compared to the vehicle control (2.45 ± 0.08 × 10^4^ AU in the 0.11 µM group, 2.4 ± 0.05 × 10^4^ AU in the 0.33 µM group, and 1.93 ± 0.09 × 10^4^ AU in the 1 µM group, compared to 2.9 ± 0.06 × 10^4^ AU in the vehicle control group, *p* < 0.05, Figure 3). These results indicate that SFN significantly decreases hyperoxia-induced oxidative stress in cultured macrophages. The SFN attenuated macrophage dysfunction of macrophages exposed to hyperoxia is associated with the decrease in hyperoxia-induced oxidative stress, confirming SFN’s antioxidant efficacy.

### 2.3. Sulforaphane Significantly Inhibits Hyperoxia-Induced Release of HMGB1 from Cultured Macrophages

To determine whether SFN-enhanced macrophage function under hyperoxic conditions is attributable to HMGB1-mediated mechanisms, we determined the extracellular levels of HMGB1 in the cell culture supernatant of RAW 264.7 cells exposed to hyperoxia. HMGB1 levels were significantly greater in the cell culture media under hyperoxic conditions compared to that of the room air control group (96.6 ± 3.9% vs. 8.1 ± 8.1%, *p* < 0.05, Figure 4). The incubation of macrophages with 0.33 µM of SFN (47.7 ± 14.7% compared to 93.1 ± 14.6% in the vehicle control group, *p* < 0.05, Figure 4), significantly decreased the levels of extracellular HMGB1 accumulated in the media. These data suggest that the prophylactic efficacy of SFN on hyperoxia-compromised macrophage functions is, in part, mediated by inhibiting HMGB1 release. These data suggest that SFN alleviates macrophage dysfunction via inhibiting the accumulation of HMGB1 in the extracellular milieu.

### 2.4. Ascorbic Acid Significantly Attenuates Hyperoxia-Induced Acute Inflammatory Lung Injury

Previously, we have shown that AA attenuates hyperoxia-compromised macrophage function and decreases the accumulation of HMGB1 in the cultured media [28]. We further tested the hypothesis that antioxidants can ameliorate HALI if they can attenuate both hyperoxia-compromised macrophage phagocytic function and hyperoxia-induced accumulation of extracellular HMGB1 in mice subjected to >98% O_2_ and treated with AA. Histological micrographs and differential cell analysis in bronchoalveolar lavage fluid (BALF) indicated that the lungs of hyperoxia-exposed mice were infiltrated by a significantly greater number of leukocytes compared to the lungs of mice exposed to room air (100 ± 5.8% versus 34.5 ± 4.2%, *p* < 0.05; Figure 5A,B). In addition, there was a significant increase in the thickness of the alveolar septa compared to that of those exposed to room air (Figure 5A). The administration of 16.6 mg/kg intraperitoneal (*i.p.*) of AA did not significantly decrease the number of airway leukocytes in mice exposed to hyperoxia compared to saline-treated controls (83.37 ± 9.543 % versus 100 ± 5.82 % in the saline-treated hyperoxia control group, *p* = 0.1274, Figure 5B). However, 50 mg/kg *i.p.* of AA produced a significant decrease in the number of leukocytes compared to saline-treated controls (60.39 ± 4.137% versus 100 ± 5.82 % in the saline-treated hyperoxia control group, *p* < 0.05; Figure 5B).

Damaged epithelial-endothelial barriers can increase the flux of cells and their debris from the vasculature into the airspaces, thereby increasing the total protein content in the airways [2,9,12]. Consequently, assessing the total protein content in BALF has been postulated to be a marker of lung injury [33]. Mice exposed to hyperoxia had significantly higher concentrations of protein in the BALF compared to mice exposed to room air (1386 ± 87.1 versus 135.6 ± 13.3 µg/mL BALF, *p* < 0.05; Figure 5C). However, hyperoxia exposed mice given 50 mg/kg *i.p.* of AA had a significantly lower total protein content in BALF compared to saline-treated hyperoxic mice (891 ± 96.3 versus 1386 ± 87.1 µg/mL BALF in saline-treated hyperoxic control mice, *p* < 0.05, Figure 5C). These results suggest that AA mitigates hyperoxia-induced lung injury by reducing hyperoxia-induced leukocyte accumulation in the airways.

### 2.5. Ascorbic Acid Significantly Attenuates Hyperoxia-Induced Oxidative Stress

Previously, we reported that AA levels are significantly decreased in macrophages exposed to hyperoxia and that supplementation with AA increases macrophage function by restoring the oxidative reduction potential (ORP) in macrophages [28]. ORP is a measure of the likelihood of the reduction of a chemical species and ROS are one of the primary biomolecules that affect the ORP [34]. We determined the ORP of BALF samples in mice exposed to hyperoxia (≥98% O_2_) for 72 h. Consistent with the data from cultured macrophages, there was a significant increase in the ORP in the airways of mice exposed to hyperoxia compared to mice exposed to room air (100 ± 0.98% versus 93.7 ± 0.5%, *p* < 0.05; Figure 6A). The *i.p.* administration of 50 mg/kg of AA significantly reduced the ORP in hyperoxia-exposed mice compared to saline-treated hyperoxic mice (94.5 ± 1.1% versus 100 ± 0.98%, *p* < 0.05; Figure 6A).

Nitric oxide can interact rapidly with superoxide to form peroxynitrite under hyperoxic conditions, which can cause cellular damage by nitration and nitrosation of enzymes, cytoskeletal proteins and other biomolecules [9]. It is likely hyperoxia, by increasing superoxide levels, can increase peroxynitrite levels, as well as other nitrogen oxide species (NOx). Therefore, we determined the levels of nitrite, a surrogate biomarker for NOx, in BALF as an indicator of the magnitude of potential nitrosative stress. Mice exposed to hyperoxia had significantly greater nitrite levels in the airways compared to mice exposed to room air (34.6 ± 5.3 µM versus 22.7 ± 2.9 µM, *p* < 0.05; Figure 6B). The administration of 50 mg/kg *i.p.* of AA significantly reduced nitrite levels in mice exposed to hyperoxia, compared to the saline control (13.4 ± 1.19 µM versus 34.6 ± 5.3 µM, *p* < 0.05; Figure 6B). Overall, these results suggest that AA is efficacious in reducing the hyperoxia-induced oxidative and nitrosative stress in the lungs of animals exposed to hyperoxia.

### 2.6. Ascorbic Acid Significantly Decreases Hyperoxia-Induced Accumulation of HMGB1 in Airways of Mice

Previously, we have shown that elevated levels of airway HMGB1 contribute to the pathogenesis of HALI [6]. In addition, AA significantly decreases hyperoxia-induced release of HMGB1 from RAW 264.7 cells into the extracellular milieu [28]. To determine whether AA attenuates HALI is accompanied with a decrease in the accumulation of HMGB1 in the airways, we measured the HMGB1 levels in the BALF of mice exposed to hyperoxia. Our results indicated that HMGB1 accumulated in the airways of mice exposed to hyperoxia and this accumulation was significantly lower in animals treated with 50 mg/kg *i.p.* of AA compared to animals treated with saline control (43.3 ± 12.2% in 50 mg/kg versus 96.7 ± 9.39%, *p* < 0.05, Figure 7). These data suggest that the prophylactic efficacy of AA on hyperoxia-induced acute inflammatory lung injury is due to a decrease in the elevated airway levels of HMGB1, which may translate to enhanced macrophage functions compromised by HMGB1.

## 3. Discussion

Acute inflammatory lung injury is a major adverse effect of mechanical ventilation, which is used with hyperoxia as the major supportive therapy for the management of patients with respiratory distress or failure [2,9,35,36]. Currently, there are no effective therapeutic interventions to address this problem. HALI caused by prolonged exposure to hyperoxia during mechanical ventilation is mediated by oxidative stress-induced cascades [36]. In this study, we report that SFN, a dietary Nrf2 inducer, effectively mitigates some characteristics associated with the pathogenesis of HALI such as restoring hyperoxia-impaired macrophage phagocytosis and reducing hyperoxia-induced HMGB1 accumulation in the extracellular milieu. Our results also indicated that AA, which enhances macrophage functions and inhibits HMGB1 release from cultured macrophages under hyperoxic conditions, significantly reduced HALI by attenuating hyperoxia-induced leukocyte infiltration in the lung tissues and decreasing the accumulation of HMGB1 in the airways. These results indicate that dietary antioxidants, which are efficacious in attenuating HMGB1-compromised macrophage phagocytosis, can also mitigate the hallmarks of HALI and, thus, may be potential treatments for ALI in patients on mechanical ventilation.

### 3.1. Antioxidants Are Effective in Mitigating Cellular Damage and HALI by Reducing Oxidative Stress in Cultured Macrophages and Animals Exposed to Hyperoxia

The involvement of ROS, particular superoxide, in HALI, is well established [9,10,37]. ROS play critical roles in the pathogenesis of the HALI by initiating and propagating significant cellular damage and pronounced proinflammatory response in lungs of animals subjected to prolonged exposure to hyperoxia [9,10,37]. Congruent with these studies, our results indicate that prolonged exposure of macrophages or mice to hyperoxia increases the levels of ROS, oxidation-reduction potential, and nitrite levels (Figure 3 and Figure 6). The increase in ROS production produces oxidative and nitrosative stress, which can result in alveolar macrophage dysfunction and loss of epithelial/endothelial integrity, and leads to excessive infiltration of leukocytes to the lung [9,10,14,16,28,36,37,38]. The hyperoxia-induced infiltration of neutrophils and monocytes into the pulmonary circulation, interstitium, and air spaces can further increase ROS levels in the lung, thereby exacerbating the resulting oxidative cell injury [9,10]. Numerous studies have been performed to investigate the efficacy of antioxidant therapies in preventing acute inflammatory injuries and ARDS. However, to date, there is no FDA approved treatment for patients receiving mechanical ventilation and hyperoxia who develop ALI/ARDS [39,40]. Recently, research has been directed toward dietary antioxidants, such as vitamin C and Nrf2-inducers. Nrf2, a member of the Cap-n-Collar family of basic leucine zipper proteins, is a transcriptional regulator of at least 1055 genes and is regarded as the master regulator of the oxidative stress response [41,42]. In an oxidative environment, Nrf2 dissociates from its cytoplasmic inhibitor Keap1, translocates to the nucleus and binds to the antioxidant response element (ARE), resulting in the expression of genes that regulate numerous cellular processes, including antioxidant and detoxifying responses [41,42,43]. Three cysteine residues in Nrf2, Cys-151, Cys-273 and Cys-288, are sensors of various electrophiles [44]. Nrf2 activation and induction of the antioxidant response is important for regulating ROS produced by the mitochondria and NADPH oxidases [42]. It has been reported that Nrf2 plays a critical role in the susceptibility to HALI in murine models [21,23]. Isothiocyanates, such as SFN, scavenge free radicals and produce anti-inflammatory efficacy by activating the Nrf2-mediated expression of antioxidant enzymes such as peroxidase 2, peroxiredoxin 1 and 6, glutathione reductase and glutamate cysteine ligase [45,46,47]. Furthermore, Cho et al. [22] reported that Nrf2 is induced by SFN in a mouse model of HALI and SFN is efficacious in decreasing the inflammatory response and markers of ALI. In this study, our results indicate that hyperoxia-induced levels of intracellular ROS in cultured macrophages can be significantly decreased by SFN in a concentration-dependent manner (Figure 3). In addition, AA directly reduces oxidative stress in the lung of mice subjected to prolonged exposure to hyperoxia (Figure 6). The reduced levels of ROS can improve macrophage function in phagocytosis (Figure 2), likely by attenuating actin oxidation in cultured macrophages [14,28].

Our findings are consistent with previous studies indicating that SFN and AA significantly decrease oxidative stress. For example, the intravenous administration of 5 mg/kg of SFN in a rabbit model of ALI significantly decreased lung injury and increased serum levels of Nrf2, suggesting that Nrf2 activation can significantly improve ALI [32]. The incubation of cultured alveolar epithelial cells with 0.5 μM of SFN significantly reduced ROS production caused by cigarette smoke through a Nrf2-dependent pathway [30]. Furthermore, an analog of AA, 3-olaurylglyceryl ascorbate, can also activate the Nrf2 pathway, decreasing intracellular ROS in normal human epidermal keratinocytes cells [48].

### 3.2. Antioxidants Are Effective in Reducing the Release of HMGB1 and Accumulation of Airway Levels of HMGB1

Wang et al. [49] reported that incubation of cultured macrophages with the endotoxin, lipopolysaccharide (LPS), significantly increased the release of nuclear HMGB1 into cell culture media. Furthermore, extracellular HMGB1 levels in the serum of subjects with sepsis is a late mediator of inflammation for septic shock mice [49]. Subsequently, the excessive accumulation of extracellular HMGB1, particularly airway and sputum HMGB1, has been reported in a variety of lung diseases [50]. For example, the levels of HMGB1 in the airways of postsurgical patients who required MV for several days were 10-fold higher compared with patients who were ventilated for 5 h [51]. We have previously shown that the prolonged exposure to hyperoxia induces the accumulation of HMGB1 in mouse airways [6]. Oxidative stress initiates the accumulation of extracellular HMGB1 in a number of lung diseases and highly elevated levels of extracellular HMGB1 are associated with ALI, multiple organ failure and poor clinical outcome in patients [50]. Furthermore, the intratracheal administration of HMGB1 can induce a concentration-dependent increase in the infiltration of interstitial/intra-alveolar neutrophils, alveolar red blood cells, levels of MPO and lung edema, similar to endotoxin-induced ALI [20], suggesting that airway HMGB1 plays a critical role in mediating inflammatory responses in ALI/ARDS [20,52]. The role of airway HMGB1 in the pathogenesis of ALI was further validated using anti-HMGB1 antibodies in a mouse model of hyperoxia-induced acute lung injury [6]. Here, we report that hyperoxia-induced accumulation of HMGB1 in mouse airways is significantly attenuated by the administration of 50 mg/kg AA and this reduction is correlated with a significant decrease in lung injury (Figure 5 and Figure 7). This hyperoxia-induced accumulation of HMGB1 in the airways is due, in part, to the release of HMGB1 from macrophages following exposure to hyperoxia (Figure 4) [6,16,28,29,53]. The addition of SFN to hyperoxic macrophage culture media significantly decreased HMGB1 release (Figure 4). Our results indicate that the dietary antioxidants, AA and SFN, inhibit hyperoxia-induced HMGB1 release from macrophages into the airways, although additional experiments will be required to determine the underlying mechanisms for antioxidant-attenuated HMGB1 release.

### 3.3. Antioxidants Can Mitigate Hyperoxia-Impaired Macrophage Function in Phagocytosis and Reduce the Accumulation of Leukocytes in Hyperoxic Lung Tissues

Prolonged exposure to hyperoxia can induce inflammatory acute lung injury, which is characterized by induced alveolar thickening, leukocyte infiltration, and an increase in total protein content in the airways, an indicator of increased alveolar and vascular permeability [36]. One major pathological feature of HALI is the presence of excessive leukocytes in the airways, as shown in Figure 5. The treatment of animals subjected to prolonged hyperoxic exposure with either *i.p.* AA or SFN significantly attenuates hyperoxia-induced accumulation of leukocytes in the airways (Figure 5) [22]. The decrease in the accumulation of leukocytes in the mouse airways, produced by the antioxidant-attenuated inflammatory responses in hyperoxic animals, could be due to 1) after binding to pneumocytes, neutrophils, and macrophages, the instilled airway HMGB1 could produce potent proinflammatory effects that induce the release of other proinflammatory cytokines, such as TNF-a, IL-1b, IL-8, MCP-1, and IL-6. These inflammatory mediators can further induce the infiltration of leukocytes into the lung as reported in both naïve mice and mice exposed to 48 h hyperoxia [6,20]; and 2) airway HMGB1 induces the accumulation of leukocytes by inhibiting the phagocytosis of apoptotic neutrophils by macrophages. The efficient phagocytosis of apoptotic neutrophils by macrophages can inhibit the production of proinflammatory cytokines such as interleukin(IL)-1beta, IL-8, granulocyte macrophages colony-stimulating factor, and tumor necrosis factor-alpha [54,55]. The failure to phagocytose apoptotic neutrophils can lead to spontaneous secondary necrosis, thereby further stimulating the production of pro-inflammatory cytokines [54,56], establishing a cycle of excessive inflammatory responses. Extracellular HMGB1 can impair in vitro macrophage function in phagocytosis [17,18]. HMGB1 inhibits macrophage phagocytosis of apoptotic neutrophils by inhibiting the interaction of phosphatidylserine on apoptotic neutrophils and the receptors for apoptotic neutrophils on the macrophage membrane [57,58]. This can sequentially inhibit the activation of the down-stream cascade for phagocytosis, including Src, FAK, Rac-1 and Erk, as well as cytoskeletal rearrangement [57,58,59]. Therefore, by attenuating hyperoxia-induced accumulation of airway HMGB1, the antioxidants AA and SFN decrease hyperoxia-induced accumulation of leukocytes into the lungs and subsequently improve the outcomes of these animals (Figure 5) [22].

The reduced levels of ROS and the decreased accumulation of leukocytes in the airways of AA and SFN treated animals may be mediated through the NF-κB pathway. AA inhibits hyperoxia-induced NF-κB activation [28], whereas SFN directly inhibits the activation of the NF-κB pathway by preventing the degradation of IκB [60]. In addition, Nrf2 and NF-κB compete with each other for the transcription coactivator, CREB binding protein (CBP), resulting in decreased NF-κB activation in cells with higher Nrf2 levels [42]. Furthermore, the inhibition of NF-κB has been shown to effectively inhibit the release and extracellular accumulation of HMGB1 in macrophages exposed to hyperoxia [38].

## 4. Materials and Methods

### 4.1. Cell Culture and Reagents

Murine macrophage-like RAW 264.7 cells (ATCC TIB-71, American Type Culture Collection, Manassas, VA, USA) were cultured in Dulbecco’s Modified Eagle Medium (DMEM, #30-2002, ATCC, Manassas, VA, USA), supplemented with 10% fetal bovine serum (#S11150H, Atlanta Biologicals, Flowery Branch, GA, USA). Cells were maintained at 37°C in 21% O_2_ and 5% CO_2_ and subcultured after reaching 80–90% confluency. Cells were scraped, centrifuged at 1100 RPM for 7 min, counted using the Countess II FL Automated Cell Counter (ThermoFisher Scientific, Waltham, MA, USA), and seeded into tissue culture plates for experimental analysis. For all experiments, cells were allowed to adhere for 6 h, and then exposed to room air (21% O_2_) or 95% O_2_ in the presence or absence of R,S-sulforaphane (#ALX-350-232-M025, Enzo Life Sciences, Farmingdale, NY, USA) (0–1 µM) or sodium L-ascorbate for 24 h. Hyperoxic exposure was performed in sealed, humidified plexiglass chambers (Billups-Rothenberg Inc., Del Mar, CA, USA) flushed with a mixture of 95% O_2_ and 5% CO_2_. The MiniOx oxygen analyzer (MSA, Medical Products, Pittsburgh, PA, USA) was used to monitor O_2_ levels.

### 4.2. Isolation and Culture of Bone Marrow Derived Macrophages

BMDM were harvested as previously described (28), with minor modifications. Femurs from 8 to 12 weeks-old male C57BL/6 mice were removed with both heads intact and transferred to a petri dish containing RPMI-1640 media supplemented with L-glutamine (#11875-093, Life Technologies, Grand Island, NY, USA), containing 10% fetal bovine serum, and 1% penicillin/streptomycin (#15140-122, Mediatech, Inc., Manassas, VA, USA). Bones were placed in 70% ethanol for 1 min, washed with sterile phosphate buffered saline (PBS) (pH = 7.4) for 1 min, and the hip and knee joints were removed. The bone marrow was flushed into a sterile 15 mL centrifuge tube by passing PBS through the knee side of the bone until all the bone marrow was flushed. The marrow was dispersed by aspirating through a 19-gauge needle twice, then centrifuged at 1500 rpm for 5 min at 4 °C. Cells were resuspended, cultured, and differentiated in RPMI-1640 media containing 10% fetal bovine serum, 1% penicillin/streptomycin, and 15% L929-cell conditioned media (ATCC CCL-1, ATCC, Manassas, VA, USA) as previously described [61]. The cells were maintained at 37 °C, 5% CO_2_, and seeded into tissue culture plates for experimental analysis.

### 4.3. Phagocytosis Assay

The phagocytosis assay was performed as previously described (29), with minor modifications. Cells were seeded in 24-well plates, allowed to adhere for 6 h, and incubated with SFN for 24 h in 95% O_2_. Subsequently, the cells were incubated at 37 °C with fluorescin isothiocyanate (FITC)-labeled minibeads (Polysciences, Warrington, PA, USA) opsonized in fetal bovine serum (FBS) at a ratio of 100:1 beads per cell. Cells were immediately placed on ice and washed with ice-cold PBS to stop phagocytosis, and extracellular beads were quenched with 0.4% trypan blue (#25-900-CI, Mediatech, Inc., Manassas, VA, USA). Cells were fixed with 3.7% paraformaldehyde (#J61984, Alfa Aeasar, Ward Hill, MA, USA) at room temperature and nuclei were stained with DAPI (#D9542, Sigma-Aldrich, St. Louis, MO, USA), and the cytoplasm stained with rhodamine phalloidin (#R415, Life Technologies, Grand Island, NY, USA) overnight to visualize the cytoskeleton. Cells were immersed in immersion oil (#16482, Cargille Laboratories, Cedar Grove, NJ, USA) and imaged using the EVOS FL Auto Imaging System (ThermoFisher Scientific, Waltham, MA, USA). Phagocytic activity was assessed by quantification of FITC minibeads per cell for approximately 200 cells per well using ImageJ Software (National Institutes of Health, Bethseda, MD, USA).

### 4.4. DCFH-DA Assay

The levels of intracellular ROS were determined using the OxiSelect Intracellular ROS Assay Kit (#STA-342, Cell Biolabs, San Diego, CA, USA), according to manufacturer’s protocol. Briefly, cells were plated in clear 96-well culture plates, allowed to adhere for 6 h, and incubated with SFN in the presence of hyperoxia for 24 h. Following incubation, the cells were washed 3 times with PBS, loaded with 100 µL 1X DCFH-DA and incubated at 37 °C for 30 min. Cells were again washed 3 times with PBS before adding 100 µL media and 100 µL 2X lysis buffer (1% phenymethanesulfonyl fluoride (PMSF) (#P7626, Sigma-Aldrich, St. Louis, MO, USA) and 1% protease inhibitor cocktail (PI) (#P8340, Sigma-Aldrich, St. Louis, MO, USA) in PBS) to each well for 5 min. Finally, 150 µL of the supernatant was transferred to a black, clear-bottomed 96-well culture plate and fluorescence was measured using the FilterMax F5 Multi-Mode Microplate Reader (Molecular Devices, San Jose, CA, USA) at 480 nm/530 nm.

### 4.5. HMGB1 Release

RAW 264.7 cells were seeded in 6-well plates and allowed to adhere for 6 h. Cells were placed in 95% O_2_ in the presence or absence of SFN (0.11, 0.33 or 1 µM) in reduced-serum Opti-Mem media for 24 h. After incubation, the cell culture media was collected and concentrated in Centricon 10K (Merck Millipore Ltd., Carrigtwohill, IRL, UK) centrifuge tubes for 40 min at 4200 RPM and the concentrated samples were used for experiments.

### 4.6. Western Blot Analysis

The protein concentration in cell lysates was determined using the Pierce bicinchoninic protein assay kit (#23225, Pierce, Rockford, IL, USA), according to manufacturer’s protocol. Forty percent of the concentrated supernatants were loaded onto 12% SDS-PAGE gels and transferred to polyvinylidene fluoride membranes (#IPVH304F0, Millipore, Burlington, MA, USA). Nonspecific binding sites were blocked by incubating membranes for 1 h at room temperature in Pierce Clear Milk Blocking Buffer (#37587, Pierce, Rockford, IL, USA), diluted in Tris-buffered saline containing 1% Tween 20 (TBST). Membranes were washed 3 times with TBST and incubated overnight at 4 °C with anti-HMGB1 rabbit polyclonal primary antibody (#D3E5, Cell Signaling Technologies, Danvers, MA, USA; 1:1000). The membranes were washed three times with TBST and incubated for 1 h with anti-rabbit horseradish peroxide-coupled secondary antibody (#NA934V, GE Healthcare, Little Chalfont, Buckinghamshire, UK; 1:5000). The membranes were again washed 3 times in TBST and immunoreactive proteins were visualized using the SuperSignal West Pico chemiluminescent substrate kit (#34577, ThermoFisher Scientific, Waltham, MA, USA), according to manufacturer’s instructions. Images were developed using the ChemiDoc MP Imaging System (BioRad Laboratories, Inc., Life Science, Hercules, CA, USA) and the bands were quantified using ImageJ Software (National Institutes of Health, Bethseda, MD, USA).

### 4.7. Animal Studies

C57BL/6 mice (male, 8 to 12 weeks old, The Jackson Laboratory, Bar Harbor, ME, USA) were used in this study, in accordance with the Institutional Animal Care and Use Committees of St. John’s University (protocol #1853 was approved on 7 April 2017). Mice were housed in a pathogen-free environment and kept in a 12 h light/dark cycle. All mice had *ad libitum* access to standard rodent food and water. Mice were randomized blindly to receive either sodium L-ascorbate (#A7631-25G, Sigma-Aldrich, St. Louis, MO, USA) (16.6 or 50 mg/kg dissolved in saline) or saline (200 µL), administered by intraperitoneal injection, every 12 h, starting 24 h after the onset of hyperoxia exposure. A 2-sample t-test at the 0.05 significance level indicated that with 12 mice per treatment group, there is an 80% power to detect an effect size of 1.2. Thus, all animal experiments were performed three separate times with 3 to 4 mice each per treatment group. Mice were exposed to hyperoxia as previously described [6]. Briefly, animals were placed in microisolator cages (Allentown Caging Equipment, Allentown, NJ, USA), kept in a Plexiglas chamber (BioSpherix, Lacona, NY, USA) and exposed to ≥98% O2 for 48 h. After 72 h of hyperoxia exposure, mice were euthanized using intraperitoneal sodium pentobarbital (120 mg/kg *i.p.*) and BALF and lung tissues were collected for further analysis. The lungs were gently lavaged twice with 1 mL of sterile, nonpyrogenic PBS solution (#21-031-CV, Mediatech, Herndon, VA, USA). BALF samples were centrifuged for 10 min at 1100 RPM and the supernatants were collected in Eppendorf tubes and stored at –80 °C.

### 4.8. Lung Histopathology

Histopathological evaluation was performed on paraffin-embedded tissues as described previously [6]. Briefly, lungs were instilled with neutral buffered 10% formalin solution through a 20-gauge angiocatheter placed in the trachea. The lungs were then immersed in neutral buffered formalin overnight and processed using conventional paraffin histology. Sections were stained with hematoxylin and eosin and examined using the Evos XL Core Microscope (Life Technologies, Grand Island, NY, USA). Microscopic examination of stained lung sections was conducted blind. Acute Lung Injury scores were assessed following the method described by Szarka R. et al. in 1997 [33]. Briefly: 0—No reaction in alveolar walls. 1—Diffuse reaction in alveolar walls, primarily neutrophilic, no thickening of alveolar walls. 2—Diffuse presence of inflammatory cells neutrophilic and mononuclear in alveolar walls with slight thickening. 3—Distinct 2–3 times thickening of the alveolar walls due to the presence of inflammatory cells. 4—Alveolar wall thickening with up to 25% of lung consolidated. 5—Alveolar wall thickening with more than 50% of lung consolidated.

### 4.9. Assay for Oxidative Stress

Oxidative stress was measured by assessing the oxidation-reduction potential (ORP) using the RedoxSYS Diagnostic System (Luoxis Diagnostics, Inc., Englewood, CO, USA). Mice were exposed to hyperoxia and treated with AA as indicated. BALF was collected after treatment. A sensor strip was used to measure the static ORP (sORP) or redox status at room temperature. Briefly, the sensor was inserted into a replaceable sensor connector module, and 30 μL of sample was applied at the sample aperture on the disposable sensor strip. After sample application, the measurements were automated by the RedoxSYS Diagnostic System (Ampio Pharmaceuticals, Littleton, CO, USA), which detected when sufficient sample equilibrium on the sensor strip was achieved.

### 4.10. Assay for Nitrogen Oxide Species (NOx)

A Griess assay kit (#G2930, Promega, Madison, WI, USA) was used to measure nitrate and nitrite species in BALF samples. All assays were performed at room temperature. A nitrate standard solution (100 µL) was serially diluted (from 100–1.6 µM) in duplicate in a 96-well, flat-bottomed, polystyrene microtiter plate. The diluting medium (PBS) was used as the standard blank. After loading the plate with samples (100 µL), the addition of vanadium (III) chloride (100 µL) to each well was rapidly followed by addition of the freshly mixed Griess reagents, sulfanilamide (50 µL) and N-(1-naphthyl)ethylenediamine dihydrochloride (50 µL). Absorbance was measured at 540 nm using a plate reader Synergy LX multi-mode reader (Biotek, Winooski, VT, USA) following a 30 min incubation period.

### 4.11. Statistical Analysis

The data are presented as the mean ± standard error of the mean (SEM) of at least three independent experiments. The data were analyzed using Student’s unpaired t test or ANOVA. The post hoc analyses were conducted using Dunnett’s or Sidak’s multiple comparison test. The *a priori* significance level was *p* > 0.05.

## 5. Conclusions

In this study, we showed that the dietary antioxidants, AA and SFN, are prophylactic in hyperoxia-induced lung injury or hyperoxia-compromised macrophage function in phagocytosis. Congruent with the findings in published studies [22,28], the results presented in this study suggest that both AA and SFN can alleviate hyperoxia-induced inflammatory acute lung injury by increasing macrophage phagocytosis via inhibiting the accumulation of extracellular HMGB1 (Figure 8). Therefore, although the two dietary antioxidants reduce oxidative stress by affecting different pathways, they share similar mechanisms in maintaining the functions of lung macrophages and the integrity of lung tissues under oxidative stress by reducing the toxic effects of extracellular HMGB1. While other antioxidants have been shown to improve hyperoxia-induced macrophage dysfunction [14,62], AA or SFN may be advantageous in the clinical setting due to their affordability, low toxicity and commercial availability [24,63]. SFN (at plasma levels up to 150 μM) was well tolerated by patients in a clinical trial, and produced no significant or toxic effects [64]. The human equivalent dose of AA at 50 mg/kg used in this study is approximately 4 mg/kg [65]. Although this is approximately 2.5 times of the suggested dose of AA in humans, significantly higher doses of AA (up to 6 g/day) have been shown to be safe in humans and effective in reducing mortality in sepsis patients [66]. Furthermore, although excessively high doses of AA have been reported to cause diarrhea, intestinal distention, and flatulence, doses of AA as high as 100 to 150 g/day were well tolerated in healthy patients [67]. Therefore, the results presented in this study suggest that the supplementation of these dietary antioxidants during oxygen therapy may prevent lung damage and preserve lung cell functions and lung tissue integrity, thus providing a promising therapeutic approach for patients receiving mechanical ventilation. Although the animals in this study were not subjected to the mechanical stretch aspect of the ventilation, oxygen management is a critical element for the intensive care of mechanically ventilated patients experiencing hypoxia. Therefore, the beneficial effects of AA and/or SFN treatment should still be applicable to patients receiving mechanical ventilation (who are frequently exposed to >60% oxygen).

## Figures and Tables

**Figure 1 ijms-21-00977-f001:**
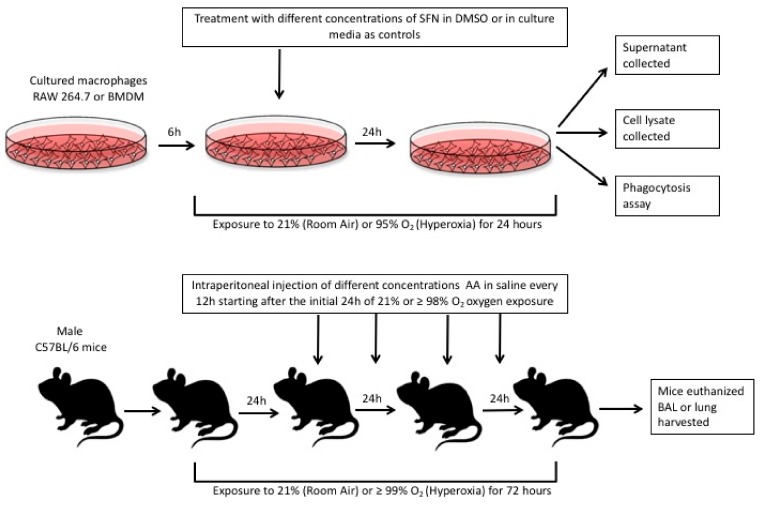
The in vitro studies were conducted in cultured RAW 264.7 macrophages and primary bone-marrow derive macrophages (BMDMs). Both types of cell cultures were seeded and allowed to adhere for 6 h. The culture media were then replaced with media containing sulforaphane (SFN) diluted in DMSO. These cells were either exposed to 24 h of 95% O_2_ (hyperoxia) or allowed to remain at room air (21% O_2_). After 24 h, different assays were conducted as described in the methods section. For the in vivo studies, male C57BL/6 mice were either exposed to > 99% O_2_ or room air (21% O_2_) for 72 h. During this 72 h period, blindly randomized groups of mice were given ascorbic acid (AA) or 0.9% saline (control group) intraperitoneally as indicated above. After 72 h, the mice were euthanized and their lung lavage fluid and lung tissue were harvested for assays as described in the methods section.

**Figure 2 ijms-21-00977-f002:**
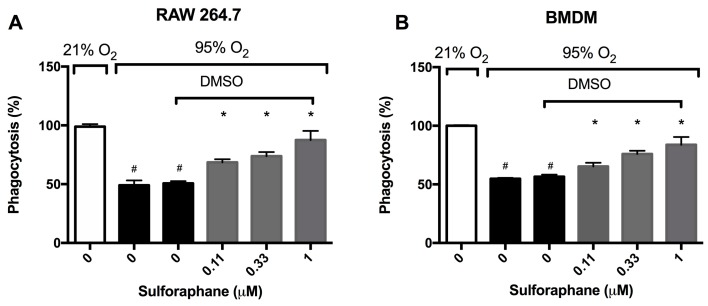
Sulforaphane (SFN) attenuates the hyperoxia-induced impairment of macrophage phagocytosis. RAW 264.7 cells (**A**) and BMDM cells (**B**) were exposed to 21% O_2_ or 95% O_2_ in the presence of increasing concentrations of SFN (diluted in DMSO as the vehicle) for 24 h and were then incubated with fluorescein isothiocyanate (FITC) labeled minibeads for 1 h. Cells were subsequently stained with DAPI and phalloidin to visualize the cells. Phagocytic activity was quantified by counting the number of minibeads in at least 200 cells per well. Data are presented as the mean ± SEM of the percentage of phagocytosed minibeads. The results were based on three independent experiments. # *p* < 0.05 compared to 21% O_2_ control group. * *p* < 0.05 compared to 0 µM SFN vehicle control group.

**Figure 3 ijms-21-00977-f003:**
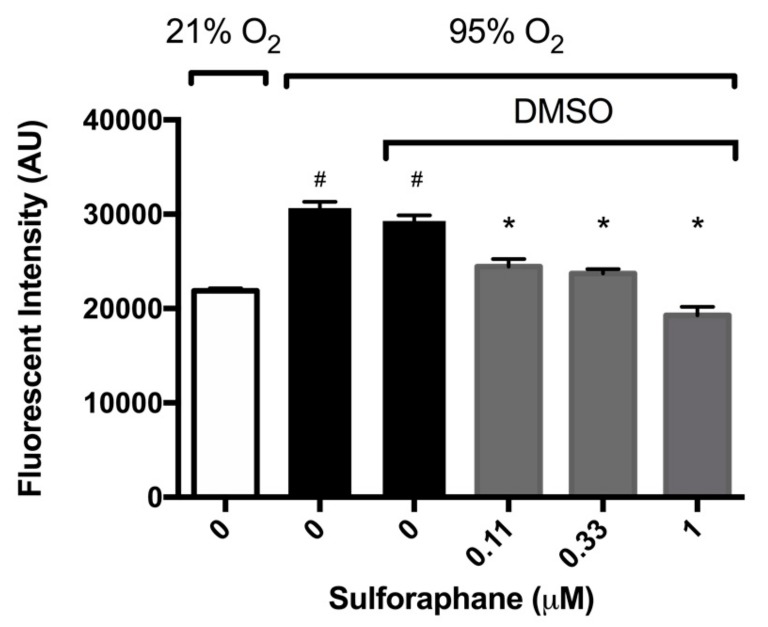
Sulforaphane (SFN) reduces intracellular ROS. RAW 264.7 cells were incubated with varying concentrations of SFN (diluted in DMSO as the vehicle) and exposed to hyperoxia for 24 h. Following hyperoxia exposure, cells were incubated with 2′,7′–dichlorofluorescin diacetate (DCFH-DA) for 30 min at 37 °C. Fluorescence was determined using a microplate reader (Excitation/Emission = 485/535 nm). Each bar represents the mean ± SEM of fluorescence for two independent experiments. # *p* < 0.05 compared to 21% O_2_ control group. * *p* < 0.05 compared to 0 µM SFN vehicle control group.

**Figure 4 ijms-21-00977-f004:**
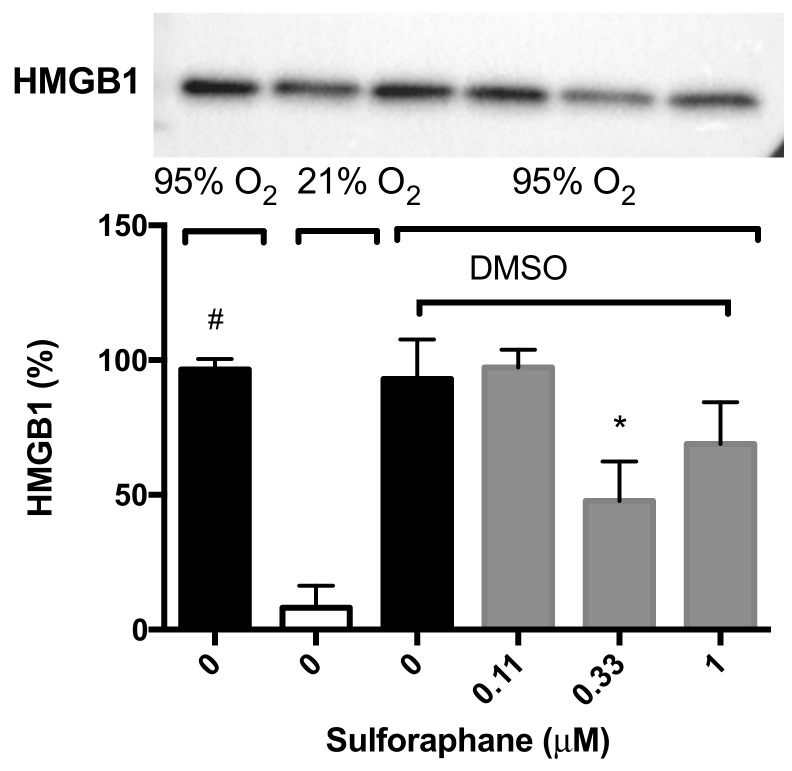
Sulforaphane (SFN) decreases the hyperoxia-induced extracellular accumulation of HMGB1 in RAW 264.7 cells. RAW 264.7 cells were exposed to 21% O_2_ (white bar) or 95% O_2_ (black bar) for 24 h in the presence of various concentrations of SFN (grey bars) (diluted in DMSO as the vehicle). HMGB1 levels in the cell culture supernatant were determined using Western blot analysis. The blot indicates the bands of HMGB1 in each group. Each value represents the mean ± SEM of at least three independent experiments. * *p* ≤ 0.05 compared to 0 µM SFN vehicle control group. # *p* ≤ 0.05 compared to 21% O_2_.

**Figure 5 ijms-21-00977-f005:**
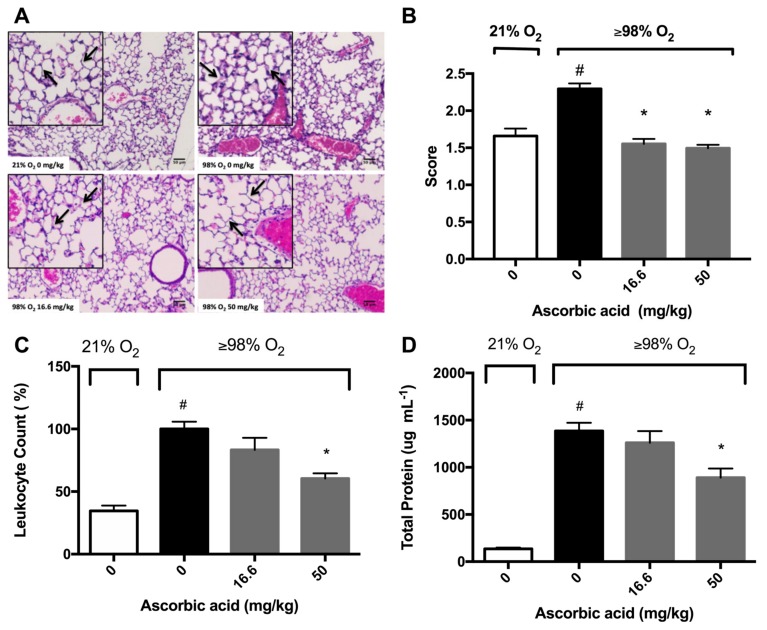
Ascorbic Acid (AA) attenuates hyperoxia-induced lung injury. Mice were exposed to ≥98% O_2_ or 21% O_2_ for 72 h and randomized to receive either AA (16.6 or 50 mg/kg *i.p.*) or saline. Mice were sacrificed and the lungs were perfused with 10% formaldehyde and harvested from the mice. Histological micrographs (originally taken at 200x magnification) with hematoxylin and eosin staining (**A**) show leukocytes (blue dots) and alveolar septa thickening (arrows) (*n* = 2–4 each group), and lungs were scored to assess inflammatory injury (**B**). In a different set of experiments, BALF was harvested and leukocyte infiltration (**C**) and protein content (**D**) were determined. The pictures are representative images from each group (*n* = 4 each group). The data represent the mean ± SEM of three independent experiments. * *p* ≤ 0.05 compared to ≥98% O_2_ 0 mg/kg control mice. # *p* ≤ 0.05 compared to 21% O_2_ control mice.

**Figure 6 ijms-21-00977-f006:**
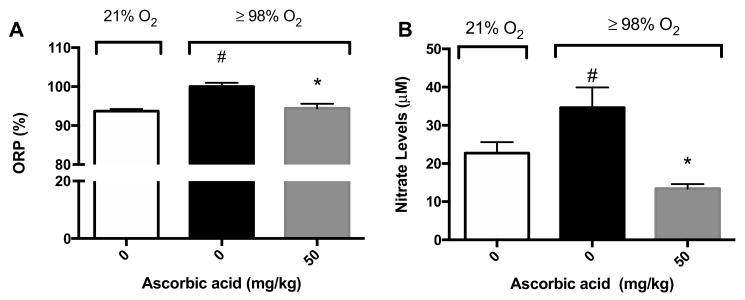
Ascorbic Acid (AA) attenuates hyperoxia-induced oxidative stress in lungs. Mice were exposed to ≥98% O_2_ or 21% O_2_ for 72 h and randomized to receive either ascorbic acid (50 mg/kg *i.p.*) or saline. Mice were sacrificed and BALF was harvested and the oxidation-reduction potential (ORP) was determined (**A**) (*n* = 8–11 each group) and nitrite levels (**B**) (*n* = 8–11 each group). The data represent the mean ± SEM of three independent experiments. * *p* ≤ 0.05 compared to 98% O_2_ control mice. # *p* ≤ 0.05 compared to 21% O_2_ control mice.

**Figure 7 ijms-21-00977-f007:**
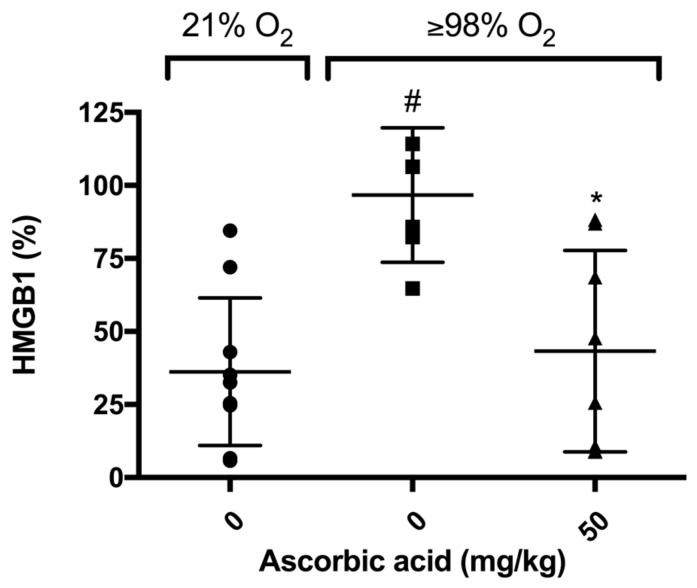
Ascorbic Acid (AA) attenuates hyperoxia-induced HMGB1 accumulation in the airways. Mice were exposed to ≥98% O_2_ or 21% O_2_ for 72 h and randomized to receive either ascorbic acid (50 mg/kg *i.p.*) or saline. Animals were sacrificed and BALF was harvested and HMGB1 levels were determined. HMGB1 levels were analyzed using Western blot analysis. Each value represents the mean ± SEM of at least three independent experiments. * *p* ≤ 0.05 compared to 98% O_2_ control mice. # *p* ≤ 0.05 compared to 21% O_2_ control mice.

**Figure 8 ijms-21-00977-f008:**
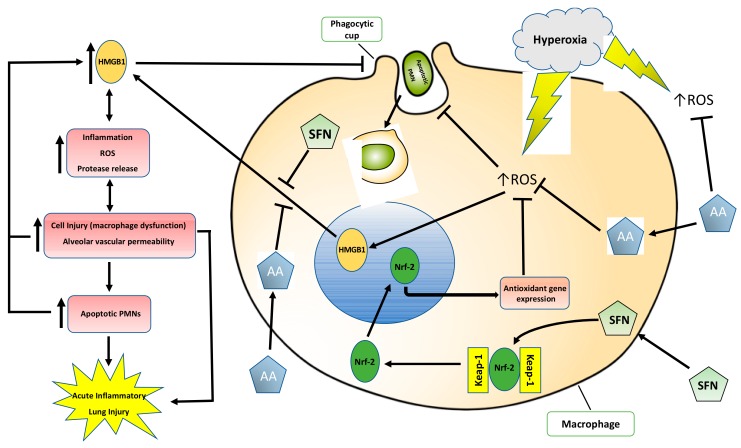
The proposed mechanism by which antioxidants ascorbic acid (AA) and sulforaphane (SFN) attenuate hyperoxia-induced lung injury and compromised macrophage function in phagocytosis. Prolonged exposure to hyperoxia increases the production of intracellular ROS and HMGB1 release, inhibiting lung macrophage phagocytosis and efferocytosis, leading to an inflammatory response that produces cell injury. The high levels of airway HMGB1 induce the infiltration of leukocytes into the airways, which further release ROS and HMGB1, contributing to a cycle of dysregulated inflammation, augmenting cell injury and leading to lung damage. Supplementation of AA or SFN significantly reduces ROS and inhibits HMGB1 release, attenuating the cycle of cell injury and ameliorating HALI.

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
