# Peer review of "Dietary Antioxidants Significantly Attenuate Hyperoxia-Induced Acute Inflammatory Lung Injury by Enhancing Macrophage Function via Reducing the Accumulation of Airway HMGB1"

_ijms, 2020, doi:10.3390/ijms21030977_

Round 1

Reviewer 1 Report

Comments:

1. Abstract

The authors should include data and appropriate statistics that support the main findings in the abstract.

2. Introduction

The introduction clearly justify why the study has been undertaken, the hypothesis and the aim of the work has been sufficiently defined. However, in order to keep the whole paper more comprehensible to the reader, the authors should consider to add a brief summary of the methods used to address every specific aim. In other words, how the authors intend to address the proposed aims should be summarized. A time-line diagram or flow chart can be useful to illustrate how the study design was carried out.

3. Methods

Please refers to the ARRIVE guidelines (www.nc3rs.org.uk/ARRIVE) for reporting animal experiments. Specifically, authors should include in the methods section full details of the number of animal for each experimental group, any sample size calculation, how animals were allocated to experimental groups, including randomisation if done.

4. Discussion

Limitation of the work should be discussed. The proposed experimental model of HALI is widely employed, but the authors should discuss the limitation of the model, including the point that the animals were not subjected to mechanical ventilation, that represent the major supportive treatment of patients with ALI.  

Author Response

1. We want to thank the reviewer for the insightful comment. We have added some of the data and statistical analysis in the abstract. 

2. We want to thank the reviewer for the insightful comment. We have added flowcharts to illustrate the experimental design and included a brief summary on the methods to address the specific aims. 

3. We thank the reviewer for the insightful comment. We have now included details in the methods section on the calculation of the number of animals. The male C57BL/6 mice were housed in ventilated cages and blindly (randomly) assigned to each group. All animal experiments were performed three separate times with 3-4 mice/group/experiment.

4. We agree with this comment. Yes, mechanical ventilation with hyperoxia does represent the major supportive treatment of patients with ALI. We have modified the discussion to indicate that the animals in this study were not subjected to the mechanical stretch aspect of the ventilation. Nonetheless, the goal of mechanical ventilation is to achieve oxygen saturation between 92-98%. Thus, oxygen management is a critical element for the intensive care of mechanically ventilated patients experiencing hypoxia. The prevention of oxygen toxicity, due to the prolonged exposure to hyperoxia, is often not regarded as a priority for the management of these patients. Therefore, our study focused on improving the oxygen toxicity-induced compromise of innate immune defenses and did not address the complications that occur to mechanical stretch. This may be a limitation of the study, but the beneficial effects of AA and/or SFN treatment should still be applicable to patients receiving mechanical ventilation (who are frequently exposed to >60% oxygen).

Reviewer 2 Report

Dear Authors,

1. The authors conducted a study in vitro and in vivo to understand the antioxidant protective effect of ascorbic acid and sulforaphane, in lung injury. Prolonged hyperoxia induce oxidative inflammatory lung injury, such it can happened during lung reperfusion or during the mechanical ventilation. The circulating protein HMGB1 was found to be a marker of hyperoxia-induced acute lung injury.

Using murine macrophage-like RAW 264.7 and bone-marrow derived macrophages cells, the authors showed that the phagocytosis is decreased in hyperoxia conditions. This decreased was decreased by sulforaphane treatment, in a dose-dependent manner, up to 1µM (a non-toxic dose). In parallel, the secretion of HMGB1 was decreased by the cells, treated with ascorbic acid and sulforaphane. Mice, exposed to 98% of O2 for 72h, were treated or not with ascorbic acid. Ascorbic acid, a Nrf2 activator, decreased lung injury and the level of inflammation. The level of the oxidative stress was decreased in a dose-dependent manner. In addition, the accumulation of HMGB1 in the airways was decreased significantly by ascorbic acid treatment. 

2. The manuscript is well written and easy to read.

3. However, I have some comments about the manuscript:

Major comments:

4. Did the authors study the intracellular ROS in the BMDM (Figure 2)? Same comment for Figure 3 and HMGB1 produced by BMDM. If not, the data should be presented in the manuscript

5. Figure 3: at 1uM, the HMGB1 seems to increase again. Could the authors perform a western blot, with a housekeeping gene, that could explain that “increase”?

6. Figure 4: the authors mentioned the use of Saline as an injection control. Does 0 means saline or is it the 98% O2 conditions only? Where are the results of this experiment? Same question for figure 5 and 6.

7. For the in vitro studies, the authors used sulforaphane, but for the in vivo studies, the authors used only ascorbic acid. Can the authors explain why they didn’t use sulforaphane for the in vivo study and ascorbic acid for the in vitro study?

8. Why do the percentage of O2 in figure 1, 2, 3 and 6 is 95% and in figure 4, 5 the percentage is 98%?

9. In conclusion, “In this study, we showed that the dietary antioxidants, AA and SFN, are prophylactic in hyperoxia-induced lung injury. Both AA and SFN significantly attenuated hyperoxia-induced inflammatory acute lung injury by enhancing macrophage phagocytosis by inhibiting the accumulation of extracellular HMGB1 (Figure 7).” Only AA was tested in vivo, on hyperoxia-induced lung injury, and sulforaphane was used in vitro. The authors should rewrite the conclusion, based on this.

10. In conclusion, the authors mentioned that the “2 dietary antioxidants reduce oxidative stress by affecting different pathways”. It could be interesting to know what their cumulative effect is, using their optimal dose, as reported in the manuscript, at least for the in vitro study.

Minor comments:

11. The symbol micro is converted in Square all over the manuscript. It should be corrected by the authors or the journal. Figure 4: what are the squares before 98%?

Author Response

1. Many thanks for your insightful and thorough reviews of this manuscript.

2. We thank the reviewer for the positive comments.

3. Below please see the point-by-point responses.

Major comments:

4. Although the analyses for intracellular ROS and HMGB1 release were not performed in BMDM, we anticipate that the results would be similar to those obtained with transformed macrophages, RAW cells. We used BMDM to confirm that transformed macrophages respond similarly as primary macrophages to hyperoxia exposure and to the treatments with antioxidants. Only the more readily available RAW cells were used to determine the mechanisms underlying SFN’s rescuing effect in hyperoxia-compromised phagocytosis

5. We thank the reviewer for this insightful suggestion. However, because we measured the levels of HMGB1 in the extracellular milieu, such as the lung lavage fluids or conditioned media, no detectable housekeeping proteins were present and therefore, could not be used as an  internal control for  the extracellular milieu. Therefore, we used the same volume of the extracellular milieu for each group to determine the release of nuclear HMGB1 into the extracellular milieu upon prolonged exposure to hyperoxia. This is a unique issue associated with many Nuclear Damage-Associated Molecular Pattern Molecules, as discussed in our recent review paper (Antioxid Redox Signal. 2019 Nov 1;31(13):954-993, The Role of HMGB1, a Nuclear Damage-Associated Molecular Pattern Molecule, in the Pathogenesis of Lung Diseases).

6. We apologize for the confusion.  We have modified the figure and figure legends to clearly indicate that “0” indicates no antioxidant was added to the vehicle as the vehicle controls for the treatment of that group of subjects (saline for AA for Figures 3-6 and DMSO for SFN for Figures 1-3). 

7. We want to thank the reviewer for the insightful comment. We did plan to determine  the effects of sulforaphane (SFN) in the in vivo model system of HALI. However, investigators from another lab reported the results of these experiments before we conducted the planned experiments. Therefore, we focused on exploring the mechanisms underlying SFN-attenuated HALI in cultured macrophages. The current study indicated that SFN attenuated hyperoxia-compromised macrophage function in phagocytosis, which plays a critical role in clearing apoptotic leukocytes in the lung.  In combination with the published study on SFN in animal models of HALI, we hypothesized that an antioxidant, which can enhance macrophage phagocytosis under hyperoxic conditions, would be  efficacious in reducing HALI. Interestingly, we also observed that the antioxidant AA attenuated hyperoxia-compromised macrophage function in phagocytosis in cultured macrophages (Patel et el., 2016). Therefore, we wanted to test this hypothesis using AA (an antioxidant) in a mouse model of HALI.  In summary, using the antioxidants AA and SFN, we confirmed our hypothesis that dietary antioxidants, which can enhance macrophage phagocytosis under hyperoxic conditions, were efficacious in reducing HALI.        

8. We thank the reviewer for noting this discrepancy. For the in vivo experiments, we could have use oxygen concentrations as high as 100% (hence the ≥98% O2). However, 5% CO2 is required to culture the macrophages. Thus, the highest concentration for the in vitro experiment is 95%.

9. We want to thank the reviewer for this insightful comment. The reviewer’s point is well taken. We have modified the conclusion as suggested. In addition, please refer to the above explanation as to why in this study, SFN was not used in the in vivo experiments, whereas AA was not used in the in vitro experiments. In brief, the reason is that the results of these experiments have been previously published. Therefore, the conclusion illustrated in Figure 7 is based on the data from this current study as well as previously published studies (Cho et al., 2019 and Patel et al., 2016).        

10. We want to thank the reviewer for this insightful suggestion. We plan to use both in vitro and in vivo model systems to determine if there is an additive or synergistic effect for the antioxidants, AA and SFN. If synergism is detected, this could be clinically significant. However, due to time restraints (about one week for the revision) and the limited access to our facilities during the holiday break at our university, we did not have the ability to perform these proposed experiments and analyze the data. However, we are confident that we will carry out these experiments in the near future and hope to report the results of these studies in 2020.    

Minor comments:

11. Thanks to the reviewer for the thorough reviewing of the figures. It is likely this problem was the result of the conversion process and thus have to be corrected by the journal.

Round 2

Reviewer 2 Report

Dear Authors,

The authors answered to all the questions. I have no more comments.

Author Response

We want to thank the reviewer for the insightful and thorough review of the manuscript.